# Factors Influencing Decision Making Regarding the Acceptance of the COVID-19 Vaccination in Egypt: A Cross-Sectional Study in an Urban, Well-Educated Sample

**DOI:** 10.3390/vaccines10010020

**Published:** 2021-12-24

**Authors:** Mohamed Elsayed, Radwa Abdullah El-Abasiri, Khaled T. Dardeer, Manar Ahmed Kamal, Mila Nu Nu Htay, Birgit Abler, Roy Rillera Marzo

**Affiliations:** 1Department of Psychiatry and Psychotherapy III, University of Ulm, Leimgrubenweg 12-14, 89075 Ulm, Germany; birgit.abler@uni-ulm.de; 2Nuffield Department of Population Health, Old Road Campus, University of Oxford Richard Doll Building, Oxford OX3 7LF, UK; radwa.el-abasiri@dph.ox.ac.uk; 3Faculty of Medicine, Cairo University, Cairo 11562, Egypt; khaledegy99@gmail.com; 4Faculty of Medicine, Benha University, Benha 13511, Egypt; manarahmedkamal37@gmail.com; 5Department of Community Medicine, Manipal University College Malaysia, Manipal Academy of Higher Education, Melaka 75150, Malaysia; drmlnnh@gmail.com; 6Department of Community Medicine, International Medical School, Management and Science University, Shah Alam 40100, Malaysia; rrmtexas@yahoo.com; 7Global Public Health, Jeffrey Cheah School of Medicine and Health Sciences, Monash University Malaysia, Petaling Jaya 46150, Malaysia

**Keywords:** vaccine, COVID-19, Egypt, intention, accepting, factors

## Abstract

Background: The COVID-19 pandemic has raised the necessity to rapidly develop safe and effective vaccines to limit the spread of infections. Meanwhile, vaccine hesitancy is a significant barrier to community vaccination strategies. Methods: An internet-based cross-sectional survey was conducted from March to April 2021 during the start of the vaccination campaigns. Results: A total of 1009 subjects participated, and the mean age (±SD) was 29.11 ± 8.2 years. Among them, 68.8% believed that vaccination is an effective method to control the spread of the disease, 81.2% indicated acceptance of the vaccine, and 87.09% reported that their doctor’s recommendation was essential for decision making. After adjusting for socioeconomic characteristics, rural residency (AOR 1.783, 95%CI: 1.256–2.531), working a part-time job (AOR 2.535, 95%CI: 1.202–5.343) or a full-time job (AOR 1.951, 95%CI: 1.056–3.604), being a student (AOR 3.516, 95%CI: 1.805–6.852) and having a partner (AOR 1.457, 95%CI: 1.062–2.00) were significant predictors for higher vaccine acceptance among the study participants. Believing in the vaccine’s efficacy showed the strongest correlation with vaccine acceptance (Spearman’s *r* = 0.309, *p* < 0.001). Conclusions: Although general vaccine acceptance is high (32.85%) in participants in our study, gender and geographic disparities were observed in the investigated urban population of young, well-educated Egyptians.

## 1. Introduction

The severe acute respiratory syndrome coronavirus 2 (SARS-CoV-2) caused the emergence of the ongoing coronavirus disease 2019 (COVID-19) pandemic [1,2,3,4]. SARS-CoV-2 spread rapidly to 223 countries [1,2]. As of 24 September 2021, the COVID-19 pandemic has caused more than 442 million infections [5]. SARS-CoV-2 can induce life-threatening complications, such as fever, dyspnoea, cough, and acute respiratory distress [6]. The COVID-19 outbreak raised the necessity to rapidly develop efficient and safe vaccines to limit the spread of this infection [7]. However, despite the serious pandemic situation, the willingness of the Egyptian people to receive a vaccine against COVID-19 is still unclear. Vaccine hesitancy is a significant barrier to vaccination campaigns, as it obstructs the process of achieving herd immunity [8]. The initial strategy used by most countries worldwide was reducing disease transmission by mask policies, social distancing, hand sanitization, travel restrictions, and lockdowns [9]. Despite the success of the measures mentioned above to slow down the disease spread, vaccination remains the most influential factor in decreasing morbidity and mortality [10]. Vaccines are reliable, cost-effective public health interventions that save millions of lives annually [11,12,13]. A worldwide race to develop a vaccine against SARS-CoV2 started after the emergence of the COVID-19 pandemic [14]. Up to now, at least ten COVID-19 vaccines were emergency authorized [15] with the hope of ending the pandemic via herd immunity [14,16]. The worldwide vaccine skepticism and hesitancy are the main obstacles to rapidly reaching herd immunity [17,18,19]. The World Health Organization (WHO) defined Vaccine Hesitancy as a “delay in acceptance or refusal of vaccination despite the availability of vaccination services” [18]. The acceptance of vaccines depends on trust in the safety and effectiveness of the vaccine and other factors, such as the political and healthcare system [20].

Until mid-September 2021, Egypt reported almost 300,000 confirmed cases of COVID-19 and over 17,000 deaths to the WHO. As of mid-September 2021, a total of nearly 13 million vaccine doses have been administered [21]. Assuming every person needs two doses, about 6.5% of the country’s population could have received the vaccine by then [22]. A web-based questionnaire in Italy of over 3000 participants from March to April 2021 showed that 91.9% were keen to receive a COVID vaccination [23]. An anonymous cross-sectional survey among over 2000 Chinese adults in March 2020 showed an acceptance rate of 91.3% to be vaccinated against COVID-19 [24]. A cross-sectional study previously investigated vaccine hesitance during the pandemic in Egyptian healthcare workers [25]. In the study conducted from 30th March to the end of April 2021, only 26% of participants were willing to be vaccinated, while 41.9% were hesitant, and 32.1% refused to be given the vaccine [26]. The current literature lacks adequate information about the potential factors that might influence vaccination against COVID-19 in the Egyptian population. Therefore, the current study investigated the factors that influence vaccine apprehension in an Egyptian community sample. This study aims to assess the COVID-19 vaccine acceptance among an Egyptian sample, highlight the factors that may influence the decision to be vaccinated against COVID-19 and raise more attention to the importance of identifying such factors. Examining the variables that affect positive and negative attitudes toward or against vaccination against COVID-19 is important for the authorities to adjust campaigns to broaden the COVID-19 vaccination quota in Egypt and address the specific concerns of hesitant risk groups.

## 2. Materials and Methods

### 2.1. Study Setting and Population

During the vaccine campaigns starting, an internet-based cross-sectional survey was conducted from March to April 2021, with the health ministry calling the population to be vaccinated. The Egyptian health ministry did not announce the vaccination quota when conducting this study. However, only medical personnel were eligible for vaccination at the time of the study. Snowball sampling was chosen as a convenient sampling method for data collection in private and business networks of colleges and their relatives. The study population was adults aged 18 to 65 years who resided in Egypt during the period mentioned above and were not eligible for vaccination at the survey time. The survey was sent online to avoid increased human contact and limit the infection rates. The structured online survey was in English and distributed through electronic mail, WhatsApp, Telegram, and other social media platforms throughout Egypt. Co-researchers and colleagues identified each respondents’ social media account through their links and networking. The survey was blinded so that the responses were not identifiable. In addition, the records were secured through password-protected files and with encryption when sending information over the internet to keep the participant’s identity confidential.

### 2.2. Ethical Considerations 

Ethical approval was granted from the Research Ethics Committee from Asia Metropolitan University (Ethics Approval Number: AMU/FOM/NF 202115). The recruitment procedure complied with the declaration of Helsinki regarding research on human subjects. All participants participated voluntarily and gave their informed consent before filling the survey.

### 2.3. Study Tool

Data were collected through a structured online questionnaire (see Appendix A). The questionnaire has two parts: Section A: Sociodemographic data (age, gender, place of residency, race, educational level, occupational status, marital status, and family income). Section B: The acceptability and factors influencing the acceptability of COVID-19 vaccination, which consists of six closed-ended questions. The first and the second questions asked if the respondent believes that the COVID-19 vaccination is an effective way to prevent and control COVID-19 and if they will accept a vaccine when it is successfully developed and approved. The following three questions inquired about the factors affecting the vaccine’s acceptability, such as method, frequency, distance to the vaccine sites, doctor’s recommendation, and price. The last question asked subjects to check (yes or no) if the participant would accept the vaccine as soon as available or wait until further testing was performed.

A group of expert panelists in Egypt, including psychiatrists, clinical psychologists, physicians, specialists, pharmacists, clinicians, and public health experts, translated and culturally validated the questionnaire into Arabic (but the language later used during the data collection was English).

### 2.4. Sample Size and Sampling 

Power analysis on this sample suggested a minimal sample size of 1000 participants to detect a meaningful impact size of *δ* = 0.2 (*α* = 0.05; two-tailed). Hence, 1009 responses were received. 

The questionnaire was pre-tested in a small sample of participants (*n* = 50) to confirm the practicability and clarity of questions. For content validity, the questionnaire was sent to three experts for revision, and their comments were taken in our consideration. For reliability, Cronbach’s alpha was calculated based on the first 50 responses from the questionnaire, and it was equal to 0.673. Cronbach’s alpha was questionable, and the value was considered high reliability and acceptable.

### 2.5. Statistical Analysis

SPSS v25.0 software was used for the data analysis and reliability testing. Continuous variables were presented as mean ± standard deviation (SD), while categorical variables were presented as number and percentage. The primary outcome was rapid acceptance or hesitancy toward COVID-19 vaccination as soon as it was available (response to question 6). Responses to questions 1 to 5 regarding the participants’ personal beliefs about the COVID-19 vaccination were treated as secondary outcome variables. Binary logistic regression was conducted to analyze the predictive factors for vaccine acceptance among population characteristics. Univariate and multivariate analyses were conducted to obtain the odds ratios (OR) and adjusted odds ratios (AOR) with their 95% confidence interval. Chi-square (chi x^2^) and partial rank correlation tests were conducted after controlling for potential confounders to correlate the beliefs regarding the COVID-19 vaccine (Questions 1 to 5) and the acceptance of vaccination among study participants (Question 6). Spearman’s *r* with correspondent *p* values were calculated. The *p*-values less than 0.05 were considered statistically significant. The correlation coefficient of ±0.1 was regarded as a weak correlation, ±0.3 as a moderate correlation and ±0.5 as a high correlation. The chi x^2^ of 0.1 was considered a weak association, 0.3 as a moderate association and 0.5 as a high association. When the results of OR equal one, this means that the odds of the main category equaling the odds of the reference category; when the results of OR are less than one, this means that the odds of the main category are less than the odds of the reference category; and when the results of OR are more than one, this means that the odds of the main category are more than the odds of the reference category. 

## 3. Results

### 3.1. Population Characteristics

A total of 1009 subjects responded to the survey. Forty-one participants did not consent to be included, leaving 968 individuals answering the questionnaire. Thus, the response rate was 95.9%. The mean age of included participants was 29.11 ± 8.2 years, and two-thirds (61.67%) were female. The majority were below 30 years old (69.73%) and reported urban places of residence (78.31%). Almost all of the participants were Egyptian (95.14%). Two-thirds (62.5%) worked a full-time job, and over half (53.10%) reported an annual income of more than EGP 10,000. Two-thirds (62.71%) were single with an education level of post-secondary (51.03%) or higher (45.76%). 

In this concern, it is worth mentioning that the Egyptian population has a median age of 24,7 years [27], 42.78% of the Egyptian´s total population lives in urban areas and cities [28], the average annual income per household in Egypt is about EGP 59.7 thousand [29]. Thus, the representativity of the sample investigated might be limited, particularly concerning the overrepresentation of urban residency and higher education in this investigation.

A third (32.85%) of participants responded that they would be willing to accept the COVID-19 vaccine as soon as possible (Question six: YES). In comparison, the rest (67.15%) stated that they preferred to delay vaccination until the safety of the vaccines were confirmed (Question six: NO, Table 1).

### 3.2. Beliefs about the Vaccination among Study Participants

Two-thirds (68.8%) of the participants believed the COVID-19 vaccine to be an effective method of controlling the spread of the disease (Question one). The majority (81.2%) reported acceptance of the vaccine if the approval for listing was made (Question two). Vaccine convenience was stated as necessary by 85.54% of the study participants (Question three). A doctor’s recommendation for vaccination was considered essential to facilitating decision making by 87.09% of respondents (Question four). The vaccination price was a concern in 74.69% of the study participants (Question five Table 2, Figure 1).

### 3.3. Predictors for Vaccine Acceptance among Population Characteristics

Univariate analysis showed that rural residency, having a part-time or full-time job, and being a student were predictors for vaccine acceptance as assessed by Question six. Conversely, being a female was a significant factor for vaccine hesitancy (Question six: NO, delay vaccination until confirming the vaccine safety). After adjusting for socioeconomic characteristics, rural residency (AOR 1.783, 95%CI: 1.256–2.531, *p* = 0.001), working a part-time job (AOR 2.535, 95%CI: 1.202–5.343 *p* = 0.015) or a full time job (AOR 1.951, 95%CI: 1.056–3.604, *p* = 0.033), being a student (AOR 3.516, 95%CI: 1.805–6.852, *p* < 0.001) and having a partner (AOR 1.457, 95%CI: 1.062–2, *p* = 0.02) were significant predictors for higher vaccine acceptance among study participants. On the other hand, being a female (AOR 0.36, 95%CI: 0.27–0.481, *p* < 0.001) remained a predictive factor for vaccine hesitancy (Table 3, Figure 2, Figure 3 and Figure 4).

### 3.4. Correlation between Beliefs about the Vaccine and Acceptance towards Vaccination

After controlling for the significant variables in the binary logistic regression model, a partial rank correlation test was conducted. Believing in the efficacy of the vaccine in controlling COVID-19 (Question one) and the development and approval of the listing of the COVID-19 vaccination (Question two) showed the strongest correlation with vaccine acceptance among the study participants (Spearman’s *r* = 0.309 and 0.277, *p* < 0.001, respectively). Convenience issues regarding vaccination (Question three) showed a weak but significant correlation with vaccine acceptance (Spearman’s *r* = 0.078, *p* = 0.016). The doctor’s recommendation and the price of the vaccine did not affect the decision regarding vaccination (Question five) among study participants (Spearman’s *r* = 0.061 and 0.021, *p* = 0.058 and 0.518, respectively) (Table 4).

### 3.5. Correlation between Population Characteristics and Acceptance towards Vaccination

Chi-square and bivariate Spearman’s correlation tests showed that females were less likely to accept COVID-19 vaccination than males (X^2^ = 51.78, Spearman’s *r* = −0.231, *p* < 0.001), people living in rural areas were more accepting toward vaccination (X^2^ = 7.07, Spearman’s *r* = 0.085, *p* = 0.008), and students and people with a job were more accepting of vaccination (X^2^ = 17.15, Spearman’s *r* = 0.102, *p* = 0.002). On the other hand, different age groups, nationality, education, monthly income showed no significant correlations with vaccine acceptance (Table 5).

## 4. Discussion

In 2019, the WHO identified ten significant threats to global health, including vaccine hesitancy and the risk of a pandemic [26]. Africa has now embarked on the biggest immunization drive in history to administer a COVID-19 vaccine to 60% of the population by June 2022 through the WHO-led COVAX facility [30]. This study assessed acceptance and attitudes regarding the COVID-19 vaccine in a sample of young to middle-aged well-educated adult Egyptians with an income of twice the average income and a slight overrepresentation of urban residency.

With a mean age of 29.11 years, the participants included in our study were younger than those investigated regarding vaccine hesitancy and attitudes toward COVID-19 vaccination in Egypt by El-Sokkary et al. [25] (37.6 years), similar to Omar and Hani [31] (29.35 years), and older than the medical students assessed by Saiedet al. [32] (20.24 years). The proportion of female participants was increased (61.67%), which is in line with similar Egyptian studies (77.6%), (65.2%), (81.30%), and (63.4%) [31,33,34]. In contrast, Omar and Hani [31] studied a sample in which more than half of the participants (58.8%) were males. As in our study, Omar and Hani [31] included more urban participants than rural participants (54.3%). Marital status in our sample was mostly single (62.71%), as in Omar and Hani (50%) [31] and different from El-Sokkary et al. (10%) [31].

In the present study, from March/April 2021, only a third (32.85%) of participants were willing to accept the COVID-19 vaccine rapidly. This was similar to the previous report by El-Sokkary et al., targeting Egyptian healthcare workers in January 2021, in which 26.0% of the participants mentioned were willing to take the vaccine [31]. Fares et al. targeted healthcare workers in Egypt from December 2020 to January 2021. Here, 21% of the participants were willing to take the vaccine [33]. Saied et al. targeted medical students at two Egyptian universities in January 2021, their acceptance rate to take the vaccine was 34.9% [32], and Omar and Hani targeted Egyptian citizens living in Egypt aged 18 years and older from January to March 2021; in which 25% mentioned being willing to take the vaccine [31]. According to our results, students have a higher vaccine acceptance rate (40.1%), while Omar and Hani, who sampled earlier, found only a 12.8% vaccine acceptance rate, even among students [31]. In comparison, in other countries, the acceptance of COVID-19 vaccination during the COVID-19 pandemic was 91.3% among Chinese adults, even in March 2020 [24], 67% among adults in the United States in May 2020 [34], 91.7% among healthcare workers in Germany in February 2021 [35] and ranged from 71 to 78% among Australian adults in July–September 2020. In Italy, the vaccine acceptance rate among undergraduate students was very high (91.9%) [35].

According to our results, 79 unemployed participants showed a relatively high rate of vaccine hesitancy (84%), compared to the Omar and Hani study that had only 73 (7.2%) of not working/housewives [31].

Our findings show that the potential factors associated with higher vaccine acceptance were rural residency, living with a partner, and being a secondary school student, which is similar to a report by Omar and Hani [31]. In our study, the vaccine price was not identified as a factor affecting the decision making regarding vaccination among participants. In contrast, the potential vaccine price affected the decision making in a previous study [31].

The Limitation and Strengths

Our study is the only study that collects data about vaccine hesitancy and acceptance during the third wave of the pandemic in Egypt, as Elgendy and Abdelrahim collected data in May 2021 [36]. The respondents were recruited with a non-random convenience sampling method in this study. There might be a limitation in generalizing the findings. The questionnaire was distributed in English. Therefore, there is a potential bias of missing the respondents with low English fluency. A substantial limitation of our study is that we did not know whether any of the participants had been vaccinated before the study. Furthermore, we did not know whether the participants had suffered from COVID-19 already. We believe that both factors could have influenced the results.

Our sample was smaller than the only two previous studies [32,33], while it was larger in sample size than almost all of the last Egyptian studies [31,33,36,37,38]. The drawbacks are that the questionnaire was administered online and in English, which might have caused some selection bias.

## 5. Conclusions

In summary, we conclude that there was a high rate of COVID-19 vaccine acceptance among the participants in our study compared with other Egyptian studies, despite the gender and geographic disparities observed in vaccine acceptance. Therefore, vaccination strategies should deliver adequate information to specific groups to achieve high vaccination coverage in Egypt.

## Figures and Tables

**Figure 1 vaccines-10-00020-f001:**
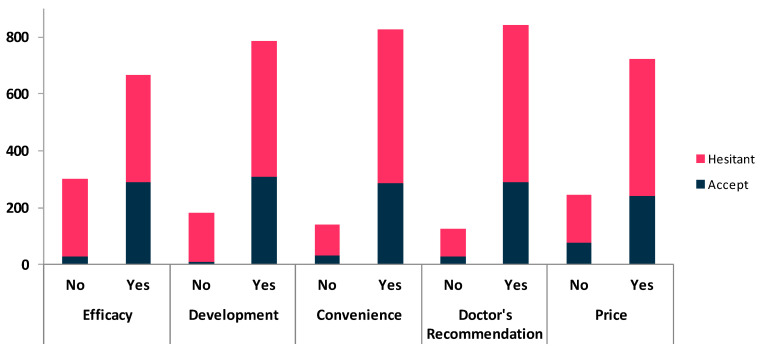
Beliefs about the vaccine and vaccination acceptance among study participants.

**Figure 2 vaccines-10-00020-f002:**
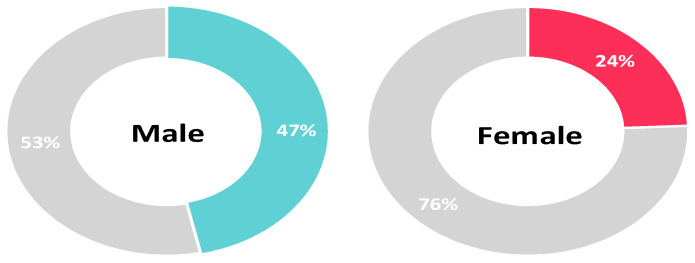
Vaccine acceptance and gender (colored section: percentage of participants that would accept the vaccine).

**Figure 3 vaccines-10-00020-f003:**
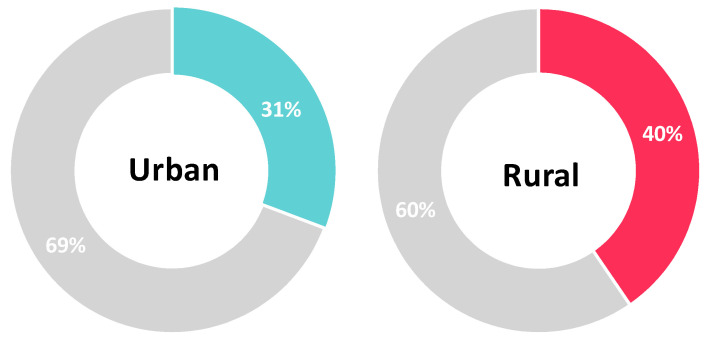
Vaccine acceptance and residency (colored section: percentage of participants that would accept the vaccine).

**Figure 4 vaccines-10-00020-f004:**
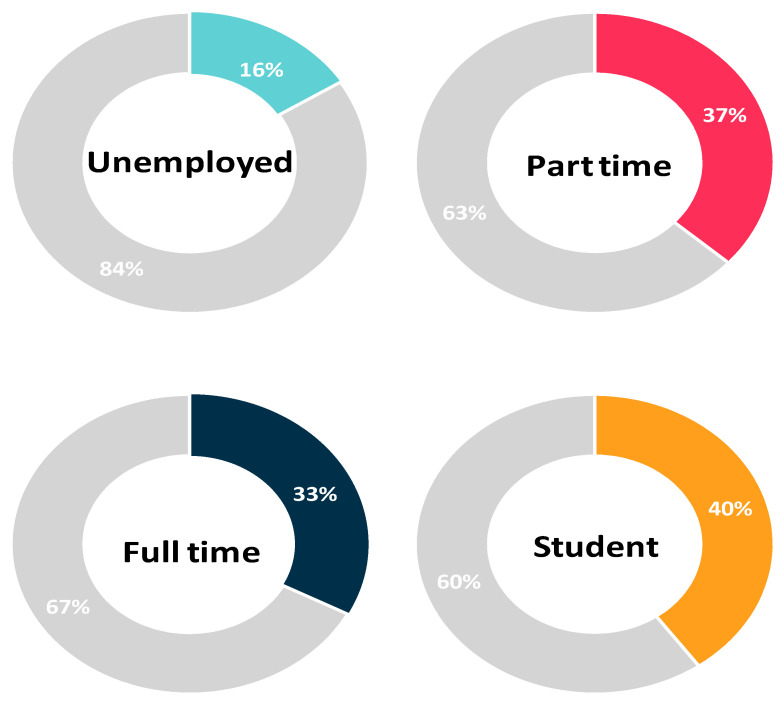
Vaccine acceptance and occupational status (colored section: percentage of participants that would accept the vaccine).

**Table 1 vaccines-10-00020-t001:** Baseline Demographic Characteristics of the Participants (*n* = 968).

Characteristic	No.	%
Age	968	
Mean ± SD	29.11 ± 8.2	
Age Groups	968	
Below 30	675	69.73
30–59	281	29.03
Over 60	12	1.24
Gender	968	
Male	371	38.33
Female	597	61.67
Residency	968	
Urban	758	78.31
Rural	210	21.69
Race	968	
Egyptian	921	95.14
Non Egyptian	47	4.86
Education	968	
Secondary or below	31	3.20
Post-Secondary	494	51.03
Tertiary	443	45.76
Occupation	968	
Unemployed	94	9.71
Part-time	82	8.47
Full time	605	62.50
Student	187	19.32
Marital status	968	
Single	607	62.71
With partner	361	37.29
Monthly Income	968	
Less than 5 k	122	12.60
5–10 k	332	34.30
More than 10 k	514	53.10
Vaccine acceptance		
I will accept the vaccine as soon as it becomes available	318	32.8
Delay vaccination until I have confirmed the safety of the vaccine	650	67.2

**Table 2 vaccines-10-00020-t002:** Beliefs about the Vaccination among Study Participants (*n* = 968).

Question Item	No		Yes	
	No.	%	No.	%
COVID-19 vaccination is an effective way to prevent and control COVID-19.	302	31.20	666	68.80
I would like to accept vaccination if the COVID-19 vaccine is successfully developed and approved for listing in the future.	182	18.80	786	81.20
Vaccine convenience (vaccination method, frequency, and distance to vaccination sites) is an important factor in vaccination decision making.	140	14.46	828	85.54
Doctor’s recommendation is an important factor in vaccination decision making.	125	12.91	843	87.09
Vaccine price is an important factor in vaccination decision making.	245	25.31	723	74.69

**Table 3 vaccines-10-00020-t003:** Predictors for Vaccine Acceptance among Population Characteristics (*n* = 968). Bold type provide easier distinction between different Items.

Characteristics	Hesitant		Accept		Unadjusted Analysis				Adjusted Analysis *			
	*n*	%	*n*	%	*p*	ORs	95	CI	*p*	ORs	95	CI
**Age Groups**	**650**		**318**									
Below 30	456	67.6	219	32.4	Ref							
30–59	186	66.2	95	33.8	0.683	1.063	0.792	1.428	0.746	0.944	0.665	1.339
Over 60	8	66.7	4	33.3	0.948	1.041	0.310	3.495	0.965	1.030	0.272	3.904
**Gender**	**650**		**318**									
Male	198	53.4	173	46.6	Ref							
Female	452	75.7	145	24.3	**<0.001**	**0.367**	**0.278**	**0.484**	**<0.001**	**0.360**	**0.270**	**0.481**
**Residency**	**650**		**318**									
Urban	525	69.3	233	30.7	Ref							
Rural	125	59.5	85	40.5	**0.008**	**1.532**	**1.117**	**2.101**	**0.001**	**1.783**	**1.256**	**2.531**
**Nationality**	**650**		**318**									
Egyptian	621	67.4	300	32.6	Ref							
Non-Egyptian	29	61.7	18	38.3	0.416	1.285	0.702	2.351	0.766	1.104	0.576	2.114
**Education**	**650**		**318**									
Secondary or below	22	71	9	29.0	Ref							
Post-Secondary	336	68	158	32.0	0.732	1.149	0.517	2.554	0.656	1.222	0.505	2.958
Tertiary	292	65.9	151	34.1	0.566	1.264	0.568	2.813	0.414	1.466	0.586	3.666
**Occupation**	**650**		**318**									
Unemployed	79	84	15	16.0	Ref							
Part time	52	63.4	30	36.6	**0.002**	**3.038**	**1.491**	**6.191**	**0.015**	**2.535**	**1.202**	**5.343**
Full time	407	67.3	198	32.7	**0.001**	**2.562**	**1.438**	**4.565**	**0.033**	**1.951**	**1.056**	**3.604**
Student	112	59.9	75	40.1	**<0.001**	**3.527**	**1.888**	**6.587**	**<0.001**	**3.516**	**1.805**	**6.852**
**Marital status**	**650**		**318**									
Single	414	68.2	193	31.8	Ref							
With partner	236	65.4	125	34.6	0.365	1.136	0.862	1.497	**0.020**	**1.457**	**1.062**	**2.000**
**Monthly Income**	**650**		**318**									
Less than 5k	87	71.3	35	28.7	Ref							
5k–10k	222	66.9	110	33.1	0.368	1.232	0.782	1.939	0.130	1.466	0.894	2.404
More than 10k	341	67.6	173	33.7	0.294	1.261	0.818	1.944	0.075	1.553	0.956	2.522

* Adjusted for Age Groups, Gender, Residency, Nationality, Education, Occupation, Marital Status and Monthly income.

**Table 4 vaccines-10-00020-t004:** Correlation between Beliefs about the vaccine and Acceptance towards Vaccination (*n* = 968). Bold type provide easier distinction between different Items.

Belief about Vaccine	Hesitant		Accept		Chi^2^	*p*	Spearman’s *r*	*p*
	*n*	%	*n*	%				
**Efficacy**	**650**		**318**					
No	272	90.1	30	9.9	**104.508**	**<0.001**	**0.309**	**<0.001**
Yes	378	56.8	288	43.2				
**Development**	**650**		**318**					
No	171	94.0	11	6.0	**73.02**	**<0.001**	**0.277**	**<0.001**
Yes	479	60.9	307	39.1				
**Convenience**	**650**		**318**					
No	107	76.4	33	23.6	**6.389**	**0.011**	**0.078**	**0.016**
Yes	543	65.6	285	34.4				
**Doctor’s Recommendation**	**650**		**318**					
No	95	76.0	30	24.0	5.098	0.024	0.061	0.058
Yes	555	65.8	288	34.2				
**Price**	**650**		**318**					
No	169	69.0	76	31.0	0.498	0.480	0.021	0.518
Yes	481	66.5	242	33.5				

Chi^2^: Chi-squared test for independence.

**Table 5 vaccines-10-00020-t005:** Correlation between Population characteristics and Acceptance towards Vaccination (*n* = 968). Bold type provide easier distinction between different Items.

Characteristics	Hesitant		Accept		Chi^2^	*p*	Spearman’s *r*	*p*
	*n*	%	*n*	%				
**Age Groups**	**650**		**318**					
Below 30	456	67.6	219	32.4	0.168	0.919	0.013	0.685
30–59	186	66.2	95	33.8				
Over 60	8	66.7	4	33.3				
**Gender**	**650**		**318**					
Male	198	53.4	173	46.6	**51.779**	**0.000**	**-0.231**	**<0.001**
Female	452	75.7	145	24.3				
**Residency**	**650**		**318**					
Urban	525	69.3	233	30.7	**7.068**	**0.008**	**0.085**	**0.008**
Rural	125	59.5	85	40.5				
**Nationality**	**650**		**318**					
Egyptian	621	67.4	300	32.6	0.664	0.415	0.026	0.416
Non-Egyptian	29	61.7	18	38.3				
**Education**	**650**		**318**					
Secondary or below	22	71	9	29	0.68	0.712	0.026	0.424
Post-Secondary	336	68	158	32				
Tertiary	292	65.9	151	34.1				
**Occupation**	**650**		**318**					
Unemployed	79	84	15	16	**17.147**	**0.001**	**0.102**	**0.002**
Part time	52	63.4	30	36.6				
Full time	407	67.3	198	32.7				
Student	112	59.9	75	40.1				
**Marital status**	**650**		**318**					
Single	414	68.2	193	31.8	0.822	0.365	0.029	0.365
With partner	236	65.4	125	34.6				
**Monthly Income**	**650**		**318**					
Less than 5k	87	71.3	35	28.7	1.122	0.571	0.025	0.429
5k–10k	222	66.9	110	33.1				
More than 10k	341	67.6	173	33.7				

Chi^2^: Chi-squared test for independence.

## Data Availability

The data at the basis of the findings of this study are available on request from the corresponding author.

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
