# Peer review of "Factors Influencing Decision Making Regarding the Acceptance of the COVID-19 Vaccination in Egypt: A Cross-Sectional Study in an Urban, Well-Educated Sample"

_vaccines, 2021, doi:10.3390/vaccines10010020_

Round 1
Reviewer 1 Report
An interesting, informative, meaningful and well written manuscript; however, there are some editing issues that the authors should consider and address. Line 41, "...limit the spread of this infection [7]." Line 53, "...vaccine availability in rapidly reaching ...". Line 59, "...reported to the World Health Organization (WHO)." Line 76 "time of conduction this study. Only ..." Line 85, "...files and with encryption when ...". Lines 96 & 97, "...prevent and control; COVID-19, as well as to accept the vaccine ...". Line 101, "...or wait until further testing was performed." Line 102, "... group of expert panelists in Egypt, including ...". Line 104, "...validated the questionnaire into Arabic ..". Line 111, "...acceptance or hesitancy towards ...". Line 112, "..as soon as it was available ...". Line 118, "...for potential confounders, in order to correlate the ...". Line 125, "...leaving 968 individuals answering the questionnaire, the ...". Line 133, "...are of 24.7 years [25], ...". Line 136, "...residency in this investigation." Line 141, "the vaccines were confirmed (Question 6...". Line 146, "...necessary by 85.54% of the study participants ...". Line 148, "...price was a concern in 74.69% of the study participants ...". Line 154, "...vaccination until confirming the vaccine ...". Line 170, "...not affect the decision regarding vaccination ...". Line 247, "...60% population covered with the COVID-19 ..". Line 255, "...than the medical students assessed by Saied ...". Line 257, "Egyptian studies (77.6%), (65.2%) ...". Lines 260 & 261, "...as in Omar & Hani study (50%) [30] and different from El-Sokkary et al. (10%)[29]." Lines 263 & 264, "...to the previous report by El-Sokkary et al., targeting Egyptian healthcare ...". Lines 269 & 270, "...18 years and older from January to March 2021, in which 25% mentioned being ...". Line 280, "...working/housewives) with a nearly similar sample ..". Lines 282 & 283, "...school student, which is similar to a report by OMar & Hani [30]." Line 293, "...COVID-19 already. We believe that both factors could have influenced the results." Lines 295 & 296, "...while it was a larger sample size than almost all of the last Egyptian studies [29], [32], ...".
Author Response
On behalf of all authors, we thank the reviewers for their comments and suggestions for improving the quality of the manuscript. As requested, in the following we will address all suggestions made by the referees' item by item. We trust that the changes made to the content and structure of the manuscript have substantially improved it and we hope that it now qualifies for publication in your estimated journal in the revised form.
#Reviewer 1:
- An interesting, informative, meaningful, and well-written manuscript; however, there are some editing issues that the authors should consider and address. Line 41, “...limit the spread of this infection [7].” Line 53, “...vaccine availability in rapidly reaching ...”. Line 59, “...reported to the World Health Organization (WHO).” Line 76 “time of conduction this study. Only ...” Line 85, “...files and with encryption when ...”. Lines 96 & 97, “...prevent and control; COVID-19, as well as to accept the vaccine ...”. Line 101, “...or wait until further testing was performed.” Line 102, “... group of expert panelists in Egypt, including ...”. Line 104, “...validated the questionnaire into Arabic”. Line 111, “...acceptance or hesitancy towards ...”. Line 112, “as soon as it was available ...”. Line 118, “...for potential confounders, in order to correlate the ...”. Line 125, “...leaving 968 individuals answering the questionnaire, the ...”. Line 133, “...are of 24.7 years [25], ...”. Line 136, “...residency in this investigation.” Line 141, “the vaccines were confirmed (Question 6...”. Line 146, “...necessary by 85.54% of the study participants ...”. Line 148, “...price was a concern in 74.69% of the study participants ...”. Line 154, “...vaccination until confirming the vaccine ...”. Line 170, “...not affect the decision regarding vaccination ...”. Line 247, “...60% population covered with the COVID-19”. Line 255, “...than the medical students assessed by Saied ...”. Line 257, “Egyptian studies (77.6%), (65.2%) ...”. Lines 260 & 261, “...as in Omar & Hani study (50%) [30] and different from El-Sokkary et al. (10%)[29].” Lines 263 & 264, “...to the previous report by El-Sokkary et al., targeting Egyptian healthcare ...”. Lines 269 & 270, “...18 years and older from January to March 2021, in which 25% mentioned being ...”. Line 280, “...working/housewives) with a nearly similar sample”. Lines 282 & 283, “...school student, which is similar to a report by Omar & Hani [30].” Line 293, “...COVID-19 already. We believe that both factors could have influenced the results.” Lines 295 & 296, “...while it was a larger sample size than almost all of the last Egyptian studies [29], [32], ...”.
Answer:
We thank Reviewer #1 for the considerate comment. We have made the requested modifications.
Reviewer 2 Report
It is a pleasure to have the opportunity to review this paper. Actually, the vaccination campaign is the first method to counteract the COVID-19 pandemic; however, sufficient vaccination coverage is conditioned by the people’s acceptance of these vaccines in the general population. In this context, the paper under review is aimed at assess the COVID-19 vaccine acceptance among an Egyptian sample.
The article is interesting but it must be deeply improved.
Title: it is overstated, I suggest to better identify the object of the study and specify the target population.
Introduction: The authors should make it clear about what is the gap in the literature that is filled with this study, considering the actual epidemiology. What is the international situation regarding the acceptance of the vaccination in the adult population (refer to articles with DOI: https://doi.org/10.3390/vaccines9060638) What is the actual contribution of the study to the literature? What are the current implications of the study?
Methods: The enrolment procedure must be better specified, who was involved in the survey? How did the authors choose the way used to send the questionnaire? How did the authors avoid the selection bias? What is the reference population? What is the minimum sample? About the questionnaire, no mention to a validation process is reported. What about face validity, reliability and intelligibility?
Statistical analysis: I suggest to insert a measure of the magnitude of the effect for the comparisons. Please consider to include effect sizes.
Discussion: I also suggest expanding. Emphasize the contribution of the study to the literature. Is there a need of improving information campaigns regarding vaccine and vaccination? discuss the implications and recommendations based on previous experience in other population groups also reporting the effectiveness of the information strategy (refer to articles with DOI: https://doi.org/10.3390/vaccines9060638), also because authors report that “being a student was significant predictors for higher vaccine acceptance”.
The Authors conclude that “Therefore, vaccination strategies should deliver adequate information to specific groups to achieve high vaccination coverage in Egypt”, this is too general. How the evidence suggest to provide information to the general population?
Limits section must be improved.Author Response
On behalf of all authors, we thank the reviewers for their comments and suggestions for improving the quality of the manuscript. As requested, in the following we will address all suggestions made by the referees' item by item. We trust that the changes made to the content and structure of the manuscript have substantially improved it and we hope that it now qualifies for publication in your estimated journal in the revised form.
#Reviewer 2:
- It is a pleasure to have the opportunity to review this paper. Actually, the vaccination campaign is the first method to counteract the COVID-19 pandemic; however, sufficient vaccination coverage is conditioned by the people’s acceptance of these vaccines in the general population. In this context, the paper under review is aimed at assessing the COVID-19 vaccine acceptance among an Egyptian sample. The article is interesting, but it must be deeply improved.
Title: it is overstated, I suggest to better identify the object of the study and specify the target population.
Answer:
We thank Reviewer #2 for the kind comment. We have modified the Title as requested. The modified title is as follows: “Factors Influencing Decision-Making Regarding The Acceptance of The COVID-19 Vaccination in Egypt: A Cross-Sectional Study in an Urban, Well-Educated Sample.”
Introduction: The authors should make it clear about what is the gap in the literature that is filled with this study, considering the actual epidemiology. What is the international situation regarding the acceptance of the vaccination in the adult population (refer to articles with DOI: https://doi.org/10.3390/vaccines9060638). What is the actual contribution of the study to the literature? What are the current implications of the study?
Answer:
We thank Reviewer #2 for the kind comment. We have addressed in the revised manuscript the current gap in the literature. The gap in the literature was presented in lines from 72 to 74 as (The current literature lacks adequate information about the potential factors that might influence the vaccination against COVID-19 in the Egyptian population).
- We explained the international situation regarding the acceptance of the vaccination in the adult population by citing the Italian and Chinese studies in lines from 65 to 70 as (A web-based questionnaire in Italy of over 3000 participants from March to April 2021 showed that 91.9% were keen to receive a COVID vaccination [23]. An anonymous cross-sectional survey among over 2000 Chinese adults in March 2020 showed an acceptance rate of 91.3% to get vaccinated against COVID-19 [24]. A cross-sectional study previously investigated vaccine hesitance during the pandemic in Egyptian healthcare workers [25]).
- The actual contribution of the study to the literature and the current implications of the study have been addressed in the revised manuscript.
Methods: The enrolment procedure must be better specified, who was involved in the survey? How did the authors choose the way used to send the questionnaire? How did the authors avoid the selection bias? What is the reference population? What is the minimum sample?
Answer: We thank reviewer #2 for the kind comments.
- The enrolment procedure was explained in lines from 88 to 89 as (Snowball sampling was chosen as a convenient sampling method for data collection in private and business networks of colleges and their relatives).
- The study population and reference population were explained in lines from 89 to 91 as (The study population was adults aged 18 to 65 years who resided in Egypt during the period mentioned above and were not eligible for vaccination at the survey time).
- The way we used to send the questionnaire was presented in lines from 91 to 96 as (The survey was sent online to avoid increased human contact and limit the infection rates. The structured online survey was in English and distributed through electronic mail, WhatsApp, Telegram, and other social media platforms throughout Egypt. Co-researchers and colleagues identified each respondents’ social media account through their links and networking).
- The way we avoided selection bias was presented in lines from 96 to 99 as (The survey was blinded so that the responses were not identifiable. In addition, the records were secured through password-protected files and with encryption when sending information over the internet to keep the participant’s identity confidential).
- The minimum sample was presented in lines from 124 to 126 as (Power analysis on this sample suggested a minimal sample size of 1000 participants to detect a meaningful impact size of δ= 0.2 (α = 0.05; two-tailed). Hence, 1009 responses were received).
- About the questionnaire, no mention to a validation process is reported. What about face validity, reliability and intelligibility?
Answer:
The language validation was mentioned in lines from 119 to 122 as (A group of expert panelists in Egypt, including psychiatrists, clinical psychologists, physicians, specialists, pharmacists, clinicians, and public health experts, translated and culturally validated the questionnaire into Arabic (but the language later used during the data collection was English)).
Information regarding validity and reliability was added in lines from 127 to 132 (The questionnaire was pre-tested in a small sample of participants (n = 50) to confirm the practicability and clarity of questions. For content validity, the questionnaire was sent to three experts for revision, and their comments were taken in our consideration. For reliability, Cronbach’s alpha was calculated based on the first 50 responses from the questionnaire, and it was equal to 0.673. Therefore, Cronbach’s alpha was questionable, and the value was considered high reliability and acceptable).
- Statistical analysis: I suggest inserting a measure of the magnitude of the effect for the comparisons. Please consider including effect sizes.
Answer:
We thank Reviewer #2 for the kind comment. We have added a correlation table. Furthermore, we inserted the measure of the magnitude of the effect for the comparisons in lines from 148 to 155 as (P-values less than 0.05 were considered statistically significant. The correlation coefficient of ± 0.1 was regarded as weak correlation, ± 0.3 moderate correlation, and ± 0.5 high correlation. The chi x2 of 0.1 was considered weak association, 0.3 moderate association, and 0.5 high association. When the results of OR equal one means that odd of main category equal odd of reference category when the results of OR less than one means that odd of the main category less than odd of reference category, and the results of OR more than one means that odd of the main category more than odd of reference category).
Discussion: I also suggest expanding. Emphasize the contribution of the study to the literature. Is there a need of improving information campaigns regarding vaccine and vaccination? discuss the implications and recommendations based on previous experience in other population groups also reporting the effectiveness of the information strategy (refer to articles with DOI: https://doi.org/10.3390/vaccines9060638), also because authors report that “being a student was significant predictors for higher vaccine acceptance”. The Authors conclude that “Therefore, vaccination strategies should deliver adequate information to specific groups to achieve high vaccination coverage in Egypt”, this is too general. How the evidence suggests to provide information to the general population? Limits section must be improved.
Answer: We thank reviewer #2 for the kind comments.
- We expanded the discussion section as requested. We believe that vaccination strategies should deliver adequate information to specific groups to achieve high vaccination coverage in Egypt. We compared our results with the Egyptian and the international literature, so we cited the recommended Italian study in lines from 323 to 324 as (In Italy, the vaccine acceptance rate among undergraduate students was very high (91.9%) [36]).
- For the limitation, we updated it in lines from 334 to 346 as (Our study is the only study that collects data about vaccine hesitancy and acceptance during the third wave of the pandemic in Egypt, as Elgendy and Abdelrahim collected data in May 2021 [38]. The respondents were recruited with a non-random convenience sampling method in this study. Therefore, there might be a limitation in generalizing the findings. In addition, the survey was conducted in the English language, and most of the participants had a good knowledge of the English language. Thus, these respondents can easily access social networks, which implies a certain
selection bias. As a result, our data could not be generalized to other sectors. A substantial limitation of our study is that we did not know whether any of the participants got vaccinated before the study. Furthermore, we did not know whether the participants had suffered from COVID-19 already. We believe that both factors could have influenced the results. In addition, our sample was smaller than only two previous studies [32], [33], while it was larger in sample size than almost all of the last Egyptian studies [31], [33], [37]– [39].
Round 2
Reviewer 2 Report
The paper has been improved and it is now suitable for pubblication